# Stock Market and Sustainable Economic Growth in Nigeria

**Erasmus L Owusu**

Data Science, Nielsen, Oxford OX3 9RX, UK; erasmus.owusu@talk21.com; Tel.:+44-186-573-2537

**Abstract:** This paper examines the relationship between stock market evolution and sustainable economic growth in Nigeria. The study employs Auto-Regressive Distributed Lag (ARDL)-bounds testing approach and a combined stock market indicators index to examine the relationship. The paper finds that, in the long run, stock markets have no positive and at best mixed effect on economic growth in Nigeria. This finding supports the numerous past studies, which have reported negative/mixed or inconclusive results on the effects of stock markets on economic growth. The paper, therefore, concludes that, there is the need for increasing financial deepening and the removal of bottlenecks in the financial sectors of the economy by providing further public and institutional education on the value of stock markets for economic development.

**Keywords:** economic growth; stock market developments; ARDL-bounds testing approach; Nigeria

## 1. Introduction

Continuous and sustained mobilisation of resources is a requisite for economic growth and development. This process has long been the focus of many development economists. To put it simply, for a sustainable growth in any given economy, financial resources must be effectively and efficiently mobilised and allocated in such a way to harness the synergies between human, material and managerial resources for optimal economic output. The financial system of a country is therefore, the framework within which capital formation takes place, and the stock market is one of the vehicles through which capital can be accumulated and channeled for effective economic growth. The stock markets do this by promoting efficient capital formation and allocation, as a tool in the mobilisation and allocation of savings among the competing choices, critical for economic growth; enabling governments and industry to raise long-term funds for new projects; and acting as an efficient capital allocator based on their rate of returns and level of risk.

The purpose of this paper is to empirically investigate and provide insight into the impact of stock markets on economic growth in Nigeria (the biggest[1] economy in Africa and member of the Economic Community of West African States—ECOWAS[2]). By trying to answer the following questions: (1) Has the evolution of the stock markets in Nigeria promoted economic growth?; (2) Has the stock market in Nigeria acted as a mechanism for attracting foreign capital inflow?; and (3) How has the stock market in Nigeria facilitated the mobilisation of resources for the financing of long-term development project?

---

[1]    Both by population and national income. Nigeria accounts for about 85% of the regions' GDP (IMF, 2015 [1]).
[2]    The Economic Community of West African States is made up of 15 countries in the in the West African sub-region, namely: Benin, Burkina Faso, Cape Verde, Cote d'Ivoire, Gambia, Nigeria, Guinea-Bissau, Guinea, Liberia, Mali, Niger, Nigeria, Senegal, Sierra Leone and Togo.

In this investigation, the paper used a variety of stock market indicators combined with a constructed stock market index, which encompasses all of the relevant stock market growth indicators in Nigeria based on data from 1987 to 2014. This is because, though the Nigeria Stock Market was established in 1960, reliable and continuous data are available from 1987 onwards. The use of different measures of stock market development would provide a better insight into the potential links between stock markets development and economic growth, thus showing clearly the aspect of stock market development that is the main driver of economic growth. The determination of whether it is stock market size or stock market liquidity (value traded and turnover ratio) that is the appropriate channel through which stock markets influence economic growth is important for policy direction.

The paper also employs the recently developed Auto-Regressive Distributed Lag (ARDL)-bounds testing approach in an attempt to establish a long-term relationship between stock markets and economic growth in Nigeria. The rest of the paper is divided into five sections. Section 2 reviews the stock market development in Nigeria, while Section 3 gives an overview of the theoretical and empirical literature. In Section 4, the methodology used, the empirical analyses, as well as the discussion of the results, have been presented. Section 5 concludes the study.

## 2. Overview of Stock Market Developments in Nigeria

The Nigerian stock market was established in 1960 as the Lagos Stock Exchange. It became the Nigerian Stock Exchange (NSE) in 1977 with branches established in different parts of the country. At the end of 1999, there were six branches: Kaduna, Port Harcourt, Kano, Onitsha, Ibadan and Lagos, which also serves as the head office of the exchange. Each branch has its own trading floor. The stock exchange creates two markets where companies can raise capital. This is often referred to as the primary market. Where shares are issued to the public for the first time and shareholders can trade in shares of listed companies—this is referred to as the secondary market. In this market, shareholders buy and sell existing shares.

The NSE has witnessed tremendous evolution since its establishment in 1960. These developments can be seen in the increasing number of capital market instruments traded in the exchange, the increase in market operators, and the tremendous increase in the size of market capitalization. Some of the major factors responsible for this development, among others are: firstly, the indigenisation of the credit base objective. This was responsible for the huge investments in the second and third development loan stock issues in 1961 and 1962. Secondly, the Income Tax Management Act of 1961. Under this act, existing pension and provident funds in the country were legally required to invest at least one-third of their funds in Nigerian Government stocks, with the potential penalty of forfeiting valuable tax concession. Thirdly, the National Provident Funds Act of 1961 required pension and provident funds established after 1961 to invest at least half of their funds in stocks. Furthermore, the Insurance and Miscellaneous Provisions Act of 1964 required that at least 25 percent of all local investment of these insurance companies must be in government securities, as the Act required the insurance companies operating in Nigeria to invest locally in at least 40 percent of their premium on locally insured risks in any financial year. Finally, the Bank of Industry (formal Nigerian Industrial Development Bank) also exerts tremendous impact on the development of the stock market in Nigeria. This it has being doing by encouraging promising enterprises to incorporate as limited liability companies and then offer to take up their shares after incorporation, and finally encouraging such companies to apply at the appropriate time for stock exchange quotation (Okonkwo et al., 2014 [2]; Nigerian Stock Exchange, 2016 [3]).

## 3. Theoretical and Empirical Literature Review

Empirical investigations of the link between stock market development and economic growth have been relatively limited in African countries, especially, the West African sub-region. Theoretically, the relationship between stock market development and economic growth has been a subject of controversy. Previous studies carried out have hardly come to a unanimous conclusion on the causal

linkage between them. While some studies maintain that stock market evolution drives economic growth, others are of the view that it stifles growth. It has also been pointed out that stock market liquidity (the ease of convertibility of assets into liquid cash at a price) also plays a very important role in the process of economic growth. Stock market liquidity reduces the downside risk and costs of investing in projects that do not pay off except after a long time. With a liquid market, the initial investors do not lose access to their savings for the duration of the investment project because they can easily and quickly sell their stake in the company. Thus, liquid stock markets could ease investment in illiquid production processes that are, of course, potentially profitable, thereby improving the allocation of capital and enhancing prospects for long-term growth (McKinnon, 1973 [4]; Bencivenga, et al., 1996 [5]; Levine, 1997 [6]; Yartey and Adjasi, 2007 [7]; Ovat, 2012 [8]).

According to the endogenous growth literature, recent theoretical studies have focused on the links between endogenous growth and stock markets. Bencivenga and Smith (1991 [9]) and Levine (1991 [10]) were among the first to propose endogenous growth models to identify the channels through which financial markets affect long-term economic growth. They put emphasis on the fact that stock markets help diversify the liquidity and investment risk of agents. Additionally, it helps to attract more savings into productive investment and prevent the early withdrawal of capital invested in the long-term projects. King and Levine (1993) [11] also suggested another approach to identify the channel of transmission between financial markets and economic growth—financial markets help the functioning of efficient resource allocation. Therefore, an economy with a well-functioning financial market will have a higher productivity growth rate. In a modern economy, banks and stock markets constitute a major part of the financial system. Although they may perform different roles in the process of economic development, their uniqueness can hardly be underestimated within the framework of economic growth. Another perspective in the relationship between financial development, investments and economic growth in endogenous growth models is concerned with financial markets, savings, investments and economic growth. Available evidence indicates that the stock market does not perform the savings function very well. Thus, more recent research on the role of the stock market in an economy has emphasised the role of a developed stock market in order to enhance the efficiency of investment, in turn leading to higher economic growth. Therefore, stock markets can enhance economic growth through investment productivity rather than the savings function (Caporale et al. 2003 [12]).

Ovat (2012 [8]) investigates empirically into the relationship between stock markets in driving economic growth, with evidence from the Nigerian stock market. Utilizing several econometric techniques, such as unit root test, co-integration test and Granger causality test, he disaggregates stock market development into two components: stock market size and stock market liquidity. His findings suggest the dominance of stock market liquidity over market size and he concludes that, while there is a two-way causation between stock market liquidity and economic growth, the strength of the causality comes more from stock market liquidity, and market size is found to have little or no effect on economic growth.

Adefeso et al. (2013 [13]) examine the long-term and causal relationship between both stock market development and economic growth in Nigeria. The Vector Error Correction Model (VECM) and co-integration techniques of analysis were employed to analyze the data and draw policy inferences on annual data from 1980 to 2010. The study finds that there is a long-term relationship between stock market development as well as banking activity variables in Nigeria. They conclude that, economic growth granger causes both stock market development and banking activity in Nigeria. The study, therefore, strongly recommends that policy makers should lay emphasis on the economic growth through the appropriate regulatory and macroeconomic policies to remove all constraints to the acceleration of the sustainability of economic growth and development in Nigeria.

Owusu and Odhiambo (2014 [14]) employ the ARDL-bounds testing approach and multi-dimensional stock market development proxies to examine the relationship between stock market development and sustainable economic growth in Ghana. They find that stock market

developments have no positive effect on economic growth both in the short- and long-term. However, they also find and conclude that an increase in credit to the private sector, rather than stock market development, is the driver of the real sector economic growth in Ghana.

Okonkwo et al. (2014 [2]) looked at determining the role and contributions of the stock market to economic growth in Nigeria from using data from 1981 to 2012 and the Johansen co-integration test to estimate the long-term equilibrium relationship among the variables. They conclude that, although the stock market size remains a very important indicator in measuring the stock market impact on economic growth, their study reveals that Nigeria's stock market size, with an average of 250 listed companies, exacts significant influence on economic growth and that economic growth and stock market capitalization have no causal relationship.

## 4. Methodology and Empirical Analysis

### 4.1. Methodology

The paper specifies three models for economic growth showing the relationship between real GDP excluding oil revenue and financial services and selected stock market indicators from 1987 to 2014. All of the models include foreign direct investments, government expenditure, and credit to the private sector as independent variables. In addition to the above independent variables, the three proxies for stock market development indicators, i.e., stock market capitalisation, values of traded stocks and stock turnover, together with an index based on the combined stock market development indicators, have been included independently in all the models. The required long-term models are specified as follows:

$$\mathrm{Ln}Y_t = \phi_0 + \phi_1 FDI_t + \phi_2 \mathrm{Ln}GEXP_t + \phi_3 \mathrm{Ln}MC_t + \phi_4 \mathrm{Ln}CPS_t + \phi_5 SMDIND_t + \varepsilon_t, \tag{1}$$

$$\mathrm{Ln}Y_t = \delta_0 + \delta_1 FDI_t + \delta_2 \mathrm{Ln}GEXP_t + \delta_3 \mathrm{Ln}VT_t + \delta_4 \mathrm{Ln}CPS_t + \delta_5 SMDIND_t + \varepsilon_t, \tag{2}$$

$$\mathrm{Ln}Y_t = \theta_0 + \theta_1 FDI_t + \theta_2 \mathrm{Ln}GEXP_t + \theta_3 \mathrm{Ln}TR_t + \theta_4 \mathrm{Ln}CPS_t + \theta_5 SMDIND_t + \varepsilon_t, \tag{3}$$

where: $Y_t$ = real total GDP excluding the contributions from the oil and the financial service sectors (RGDP); $MC_t$ = real stock market capitalisation (% of GDP); $VT_t$ = real stock value traded (% of GDP); $TR_t$ = stock market turnover ratio (%); $GEXP_t$ = real government expenditure (% of GDP); $FDI_t$ = real foreign direct investments (% of GDP); $CPS_t$ = real credit to the private sector (% of GDP); $SMDIND$ = stock market development index; $\phi_0$, $\delta_0$, and $\theta_0$ are constant parameters; $\phi_i$, $\delta_i$, and $\theta_i$ are long term elasticities or coefficients; $\varepsilon_t$ = the white noise error term; and Ln = natural log operator. The data used in this paper are taken from the World Bank's World Development Indicators (World Bank, 2016 [15]) and the World Bank's African Development Indicators (World Bank, 2011 [16]). The econometric software used in this paper is Microfit 5.0 (Oxford University Press, Oxford, UK).

According to neo-classical economic thinking, capital market developments will lead to economic growth, as a result of inflow of investments from outside the liberalised economy. To test the impact of stock markets on economic growth, therefore, real GDP ($Y = RGDP$), a measure of economic activities, is modeled as a function of stock market development indicators and other macro-economic factors (Owusu and Odhiambo, 2014 [17]).

The macroeconomic factors included are as listed above. In the first place, real government expenditure (GEXP) is calculated as a ratio of GDP. This variable was included because it is expected to crowd-out private investments. This has consequences on financial deepening and hence economic growth. Barro and Sala-i-Martin (1995 [18]) argue that government expenditure does not directly affect productivity but will lead to distortions in the private sector. One can argue that government expenditure can be growth enhancing too. This is mostly the case in the developing countries, like the three selected ECOWAS countries where the bulk of investments come in the form of government expenditure. Nurudeen and Usman (2010 [19]), for example, show that government expenditures

in the transport, communication and health sectors have a positive impact on economic growth in Nigeria (Owusu, 2012 [20]).

The credit to the private sector (CPS), as the total credit extended to the private sector by the banks to the GDP, measures the level of activities and efficiency of the financial intermediation. An increase in the financial resources, especially credits, to the private sector is expected to increase private sector efficiency and production, consequently leading to economic growth (Owusu, 2012 [20]).

The other control variable is FDI, which serves as an effective means of transferring technology to the developing countries. Economists are of the view that FDI tends to foster economic growth through its effect on the level of GDP per capita, as well as its growth. Beck et al. (2000 [21]) listed three key stock market indicators in measuring size, activity, and efficiency. The ratio of stock market capitalisation (MC) to GDP, for example, measures the size of the stock market because it aggregates the value of all listed shares in the stock market. However, the size of the stock market does not provide any indication of its liquidity. To measure stock market liquidity, they used the value of stock traded to GDP variable (VT). This indicator is equal to the value of the trades of domestic stocks divided by GDP. Lack of liquidity in the stock market reduces the incentive to investment, as it diminishes the efficiency at which resources are allocated, and, hence, it affects economic growth and development. In order to capture the efficiency of the domestic stock market, they suggested the use of the Turnover Ratio (TR), which is equal to the value of trades of shares on the stock markets divided by market capitalisation (Naceur et al., 2008 [22]). Other writers, including Bencivenga et al. (1996 [5]) are also of the view that more efficient stock markets can foster better resource allocation and spur economic growth.

Finally, to account for the combined impact of stock market activities on economic growth, an index of the three proxies of stock market development (SMDIND) is included. SMDIND is a composite index of the three stock development indicators, constructed by using their growth rates, similar to Demirguc-Kunt and Levine (1996 [23]). However, in this paper, to derive the index, the paper first computes the annual growth rate for market capitalisation (MC), the ratio of total stock value traded to GDP (VT), and the turnover ratios (TR) for each year. Thereafter, a geometric average of the growth rates is taken, in order to obtain an overall index of the stock market evolution for each year. This index allows us to examine the overall effects of stock market activities on economic growth in Nigeria. Theoretically, all of the coefficients are expected to be positive in the long run.

The methodology used in this study is based on the ARDL-bounds testing approach—the unrestricted error correction model (UECM) (Pesaran et al., 2001 [24]). The approach involves two stages: In stage one, the ARDL model of interest is estimated by using the ordinary least squares (OLS) so as to determine the existence of a long-term relationship among the relevant variables. In order to test the null hypothesis of no long-term relationship among the variables in the models, a Wald *F*-test for the joint significance of the lagged levels of the variables is performed. If the *F*-statistic is above the upper critical value, the null hypothesis of long-term relationship is accepted, irrespective of the orders of integration for the time series. Conversely, if the test statistic falls below the lower critical value, then the null hypothesis cannot be accepted. However, if the statistic falls between the upper and the lower critical values, then the result is inconclusive.

Once the long-term relationship or co-integration has been established, the second stage involves the estimation of the long-term coefficients (which represent the optimum order of the variables after selection by the Akaike Information Criteria (AIC) or the Schwarz–Bayesian Criteria (SBC). A general error-correction model (ECM) is then formulated as follows:

$$\Delta \text{Ln}Y_t = c_0 + \sigma_1 \text{Ln}Y_{t-1} + \sigma_2 FDI_{t-1} + \sigma_3 \text{Ln}GEXP_{t-1} + \sigma_4 \text{Ln}MC_{t-1} + \sigma_5 \text{Ln}CPS_{t-1} +$$

$$\sigma_5 SMDIND_{t-1} + \sum_{i=1}^{p} \alpha_i \Delta \text{Ln}Y_{t-i} + \sum_{i=0}^{q} \zeta_k \Delta FDI_{t-k} + \sum_{i=0}^{q} \varphi_m \Delta \text{Ln}GEXP_{t-m} +$$

$$\sum_{i=0}^{q} \eta_n \Delta \text{Ln}MC_{t-n} + \sum_{i=0}^{q} \lambda_r \Delta \text{Ln}CPS_{t-r} + \sum_{i=0}^{q} \eta_n \Delta SMDIND_{t-v} + \varepsilon_t, \tag{4}$$

$$\Delta LnY_t = c_1 + \delta_1 LnY_{t-1} + \delta_2 FDI_{t-1} + \delta_3 LnGEXP_{t-1} + \delta_4 LnVT_{t-1} + \delta_5 LnCPS_{t-1} +$$

$$\delta_5 SMDIND_{t-1} + \sum_{i=1}^{p} \alpha_i \Delta LnY_{t-i} + \sum_{i=0}^{q} \zeta_k \Delta FDI_{t-k} + \sum_{i=0}^{q} \varphi_m \Delta LnGEXP_{t-m} +$$

$$\sum_{i=0}^{q} \eta_n \Delta LnVT_{t-n} + \sum_{i=0}^{q} \lambda_r \Delta LnCPS_{t-r} + \sum_{i=0}^{q} \eta_n \Delta SMDIND_{t-v} + \upsilon_t, \tag{5}$$

$$\Delta LnY_t = c_2 + \phi_1 LnY_{t-1} + \phi_2 FDI_{t-1} + \phi_3 LnGEXP_{t-1} + \phi_4 LnTR_{t-1} + \phi_5 LnCPS_{t-1} +$$

$$\phi_5 SMDIND_{t-1} + \sum_{i=1}^{p} \alpha_i \Delta LnY_{t-i} + \sum_{i=0}^{q} \zeta_k \Delta FDI_{t-k} + \sum_{i=0}^{q} \varphi_m \Delta LnGEXP_{t-m} +$$

$$\sum_{i=0}^{q} \eta_n \Delta LnTR_{t-n} + \sum_{i=0}^{q} \lambda_r \Delta LnCPS_{t-r} + \sum_{i=0}^{q} \eta_n \Delta SMDIND_{t-v} + \mu_t, \tag{6}$$

where: $\sigma_i$, $\delta_i$, and $\phi_i$ are the long run multipliers corresponding to long run relationships; $c_0$, $c_1$, and $c_2$ are the drifts; $\alpha_i$, $\zeta_k$, $\varphi_m$, $\eta_n$, and $\lambda_r$ are the short term coefficients; and $\varepsilon_t$, $\upsilon_t$, and $\mu_t$ = white noise errors.

The short-term effects in the above equations are captured by the coefficients of the first differenced variables in the UECM model. According to Bahmani-Oskooee and Brooks (1999 [25]), the existence of a long-term relationship does not necessarily imply that the estimated coefficients are stable. This, therefore, implies that there is the need to perform a myriad of tests diagnoses on the selected model. This involves testing of the residuals for homoscedasticity, serial correlation, etc., as well as stability tests to ensure that the estimated model is statistically robust.

The general UECM model is tested downwards sequentially, by dropping the statistically insignificant first differenced variables for each of the equations—to arrive at a "goodness-of-fit" model—using a general-to-specific strategy (Poon, 2010 [26]; Owusu, 2012 [20]). Once the co-integration relationships have been established, the long-term elasticities or coefficients can then be generated from UECM (Owusu, 2012 [20]).

## *4.2. Empirical Analysis*

### 4.2.1. Unit Root Tests for Variables

Tables 1–4 below show the results of the Augmented Dickey–Fuller (ADF) and Phillips and Peron (1988 [27]), the PP unit root tests for the relevant variables. The PP truncation lag is selected automatically on the Newey–West bandwidth.

**Table 1.** Augmented Dickey–Fuller (ADF) unit root tests for the variables in levels.

| Variable | No Trend | Result | Trend | Result |
|---|---|---|---|---|
| *LnCPS* | −2.779 | N | −3.564 ** | S |
| *FDI* | −2.931 ** | S | −4.111 ** | S |
| *LnMC* | −2.243 | N | −1.891 | N |
| *LnRGDP* | −0.259 | N | −1.269 | N |
| *SMDIND* | −3.602 ** | S | −3.932 ** | S |
| *LnGEXP* | −3.826 ** | S | −3.673 ** | S |
| *LnTR* | −1.735 | N | −0.913 | N |
| *LnVT* | −1.781 | N | −2.016 | N |

Notes: S = Stationary and N = Non-stationary. Ln is the natural log operator. ** denotes the rejection of the null hypothesis at 5% significant level.

**Table 2.** ADF unit root tests for the variables in first difference.

| Variable | No Trend | Result | Trend | Result |
|---|---|---|---|---|
| *ΔLnCPS* | −5.098 *** | S | −5.042 *** | S |
| *ΔLnVT* | −3.360 ** | S | −3.403 ** | S |
| *ΔLnMC* | −5.399 *** | S | −5.521 *** | S |
| *ΔLnRGDP* | −6.440 *** | S | −6.722 *** | S |
| *ΔLnTR* | −4.200 ** | S | −5.003 *** | S |

Notes: S = Stationary and N = Non-stationary. Δ is the difference operator and L is the natural log operator. ** and *** denote the rejection of the null hypothesis at 5% and 1% significant levels respectively.

**Table 3.** PP unit root tests for the variables in levels.

| Variable | No Trend | Result | Trend | Result |
|---|---|---|---|---|
| *LnCPS* | −2.343 | N | −2.361 | N |
| *FDI* | −3.051 ** | S | −3.482 ** | S |
| *LnMC* | −2.651 | N | −1.573 | N |
| *LnRGDP* | −0.084 | N | −1.18 | N |
| *SMDIND* | −5.088 ** | S | −5.930 ** | S |
| *LnGEXP* | −5.325 ** | S | −5.054 ** | S |
| *LnTR* | −1.73 | N | −0.963 | N |
| *LnVT* | −1.677 | N | −1.39 | N |

Notes: S = Stationary and N = Non-stationary. Ln is the natural log operator. ** denotes the rejection of the null hypothesis at 5% significant level.

**Table 4.** PP unit root tests for the variables in first difference.

| Variable | No Trend | Result | Trend | Result |
|---|---|---|---|---|
| Δ*LnCPS* | −7.534 *** | S | −7.945 *** | S |
| Δ*LnVT* | −5.515 ** | S | −6.106 *** | S |
| Δ*LnMC* | −6.195 *** | S | −8.967 *** | S |
| Δ*LnRGDP* | −7.357 *** | S | −8.238 *** | S |
| Δ*LnTR* | −5.587 ** | S | −7.383 *** | S |

Notes: S = Stationary and N = Non-stationary. Δ is the difference operator and L is the natural log operator. ** and *** denote the rejection of the null hypothesis at 5% and 1% significant levels respectively.

As can be seen from Tables 1–4, all of the variables are either I(0) or I(1)—using the Augmented Dickey–Fuller and the Phillips and Peron (PP) unit root tests. The paper, therefore, rejects the null hypothesis that the variables non-stationary.

### 4.2.2. ARDL-Bounds Test

The results of the co-integration test, based on the ARDL-bounds testing approach, are reported in Table 5. The results show that, in all of the models, the null hypothesis of no co-integration is rejected. This implies that there is a long-term co-integration relationship among the variables in all of the models in Nigeria. The long-term results of the remaining selected models are reported in Tables 4–6 below.

Table 6 shows that the coefficient of Market capitalisation (LnMC) has a significant positive effect on economic growth (LnRGDP) in the long run. It also shows that the coefficient of the index of stock market evolution has significant but unexpected negative effect on economic growth. This may suggest that, contrary to a priori expectation, in the long run, stock market developments may have net negative impact on economic growth in Nigeria. Thus, a 1% increase in the combined index of the stock market developments leads to an approximately 5.5% decrease in economic growth in Nigeria. Note the coefficient of foreign direct investment (FDI) is negative and statistically insignificant.

**Table 5.** Bounds *F*-test for co-integration for all the economic growth models.

| Dependent Variable | Functions | | | | | *F*-Test Statistics |
|---|---|---|---|---|---|---|
| | Model 1: $F_{LnY}$ (Ln*Y* | Ln*MC*, *FDI*, *SMDIND*) | | | | | 6.83 *** |
| Ln*Y* = Ln*RGDP* | Model 2: $F_{LnY}$ (Ln*Y* | Ln*VT*, *FDI*, *SMDIND*) | | | | | 4.43 ** |
| | Model 3: $F_{LnY}$ (Ln*Y* | Ln*TR*, *FDI*, *SMDIND*) | | | | | 4.61 ** |
| **Asymptotic Critical Values** | | | | | | |
| | 1% | | 5% | | 10% | |
| Pesaran et al. (2001), ([24], p. 301), Table CI(iv) Case IV | I(0) | I(1) | I(0) | I(1) | I(0) | I(0) |
| | 4.30 | 5.23 | 3.38 | 4.23 | 2.97 | 3.74 |

Note: ** and *** denote statistical significant at the 5% and 1% levels respectively.

**Table 6.** Economic growth and market capitalisation—Results of ARDL (1, 0, 1, 2) long-term model selected on AIC (Equation (1)).

| Regressor | Co-Efficient | Standard Error | T-Ratio | Probability |
|-----------|--------------|----------------|---------|-------------|
| *C* | 2.1638 | 0.177 | 0.201 | 0 |
| *LnMC* | 2.001 | 1.072 | 1.866 | 0.079 |
| *FDI* | −0.316 | 0.27 | −1.171 | 0.258 |
| *SMDIND* | −5.534 | 2.841 | −1.948 | 0.068 |

Notes: Dependent variable: Ln*Y* = Ln*RGDP*.

In Table 7, the coefficients of value of stock traded (Ln*VT*) is statistically insignificant. However, the coefficient of the combined index for stock market evolution is significant but negative. Table 8 shows that, the coefficient of stock turnover (Ln*TR*) is statistically significant and have the expected sign, while the coefficient of the combined index of stock markets evolution is statistically insignificant and have an unexpected sign. The results shown in the three tables above would suggest that in the long run, stock market developments in Nigeria has negative impact and foreign direct investments have insignificant impacts on economic growth. The results of the short-run dynamics emanating from the long-term relationships are shown in Tables 9–11.

**Table 7.** Economic Growth and Value of Stock Traded—Results of the ARDL (2, 2, 2, 2) long-term model selected on AIC (Equation (2)).

| Regressor | Co-Efficient | Standard Error | T-Ratio | Probability |
|-----------|--------------|----------------|---------|-------------|
| *C* | 2.463 | 0.476 | 5.177 | 0 |
| *LnVT* | 0.162 | 0.217 | 0.745 | 0.469 |
| *FDI* | 0.431 | 0.285 | 1.511 | 0.155 |
| *SMDIND* | −5.777 | 1.965 | −2.94 | 0.011 |

Notes: Dependent variable: Ln*Y* = Ln*RGDP*.

**Table 8.** Economic growth and stock turnover—Results of the ARDL (1, 2, 0, 4) long-term model selected on AIC (Equation (3)).

| Regressor | Co-Efficient | Standard Error | T-Ratio | Probability |
|-----------|--------------|----------------|---------|-------------|
| *C* | 2.449 | 0.021 | 11.589 | 0 |
| *LnTR* | 0.281 | 0.086 | 3.256 | 0.007 |
| *FDI* | 0.011 | 0.079 | 0.138 | 0.893 |
| *SMDIND* | −0.223 | 0.189 | −1.178 | 0.262 |

Notes: Dependent variable: Ln*Y* = Ln*RGDP*.

**Table 9.** Economic growth and market capitalisation—Results of the ARDL (1, 0, 1, 2) ECM model selected on AIC (Equation (4)).

| Regressor | Co-Efficient | Standard Error | *T*-Ratio | Probability |
|-----------|--------------|----------------|-----------|-------------|
| Δ*FDI* | −0.041 | 0.028 | 1.472 | 0.157 |
| Δ*SMDIND* | −0.335 | 0.1 | 3.364 | 0.003 |
| Δ*LnMC* | 0.273 | 0.116 | 2.353 | 0.03 |
| Δ*LnMC*$_{-1}$ | 0.291 | 0.092 | 3.156 | 0.005 |
| *Ecm(−1)* | −0.13 | 0.069 | 1.872 | 0.077 |

| | |
|---|---|
| *R*-Squared 0.638 | Residual Sum of Squares 0.499 |
| *R*-Bar-Squared 0.489 | DW-statistic 1.844 |
| S.E. of Regression 0.171 | Akaike Info. Criterion (AIC) 5.466 |
| *F*-Stat. $F_{(5,19)}$ 5.99 [0.002] | Schwarz Bayesian Criterion (SBC) 0.590 |

**Table 10.** Economic growth and value of stock traded—Results of the ARDL (2, 2, 2, 2) ECM model selected on AIC (Equation (5)).

| Regressor | Co-Efficient | Standard Error | *T*-Ratio | Probability |
|---|---|---|---|---|
| $\Delta LnRGDP_{-1}$ | −0.521 | 0.248 | −2.103 | 0.052 |
| $\Delta FDI$ | −0.064 | 0.027 | −2.638 | 0.031 |
| $\Delta FDI_{-1}$ | −0.105 | 0.041 | −2.545 | 0.022 |
| $\Delta SMDIND$ | −0.832 | 0.166 | −5.01 | 0 |
| $\Delta SMDIND_{-1}$ | 0.106 | 0.089 | 1.2 | 0.249 |
| $\Delta LnVT$ | 0.527 | 0.118 | 4.47 | 0 |
| $\Delta LnVT_{-1}$ | 0.282 | 0.143 | 1.977 | 0.066 |
| *Ecm(−1)* | −0.249 | 0.113 | −2.2 | 0.043 |
| *R*-Squared 0.826 | | Residual Sum of Squares 0.240 | | |
| *R*-Bar-Squared 0.679 | | DW-statistic 2.132 | | |
| S.E. of Regression 0.136 | | Akaike Info. Criterion (AIC) 10.623 | | |
| *F*-Stat. *F*(8,16) 7.71 [0.000] | | Schwarz Bayesian Criterion (SBC) 3.310 | | |

**Table 11.** Economic growth and stock turnover—Results of the ARDL (1, 2, 0, 4) ECM model selected on AIC (Equation (6)).

| Regressor | Co-Efficient | Standard Error | *T*-Ratio | Probability |
|---|---|---|---|---|
| $\Delta FDI$ | −0.102 | 0.029 | −3.552 | 0.003 |
| $\Delta FDI_{-1}$ | −0.044 | 0.029 | −1.552 | 0.15 |
| $\Delta SMDIND$ | −0.103 | 0.085 | −1.202 | 0.249 |
| $\Delta LnTR$ | −0.174 | 0.119 | −1.46 | 0.166 |
| $\Delta LnTR_{-1}$ | −0.273 | 0.104 | −2.614 | 0.02 |
| $\Delta LnTR_{-2}$ | −0.139 | 0.104 | −1.335 | 0.203 |
| $\Delta LnTR_{-3}$ | −0.327 | 0.09 | −3.639 | 0.003 |
| *Ecm(−1)* | −0.461 | 0.13 | −3.554 | 0.003 |
| *R*-Squared 0.821 | | Residual Sum of Squares 0.246 | | |
| *R*-Bar-Squared 0.671 | | DW-statistic 1.896 | | |
| S.E. of Regression 0.143 | | Akaike Info. Criterion (AIC) 8.536 | | |
| *F*-Stat. *F*(8,14) 6.861 [0.001] | | Schwarz Bayesian Criterion (SBC) 2.291 | | |

As can be seen from Table 9, all of the coefficients of the changes are statistically significant apart from the coefficient of the change in the FDI. The coefficient of the error correction model, ECM(−1), is found to be statistically significant at 10% level and has the expected negative sign. This confirms the existence of a long-term relationship between the variables. This may suggest that a shock to the model is totally adjusted at 13% per annum. Thus, a shock will take about seven years and eight months to fully recover. In Table 10, all of the coefficients of the changes are statistically significant apart from the coefficient of the change in the lag of the combined stock market index (SMDIND). The coefficient of the error correction model, ECM(−1), is also found to be statistically significant at 5% level and has the expected negative sign. This confirms the existence of a long-term relationship between the variables. This may suggest that a shock to the model is totally adjusted at about 25% per annum. Table 11 also shows that there is a long-term relationship between Real GDP, foreign direct investments, stock turnover ratio and a combined index of stock market evolution. This is confirmed by the negative coefficient of ECM(−1), and it means that any shock is cleared at the rate of about 46% per year. As with the long-term modes, the results short-term dynamic models show that stock market developments and foreign direct investments in Nigeria have mixed but negative impacts on economic growth.

Finally, the regression for the underlying ARDL models fits very well, and they pass all the diagnostic tests against serial correlation, functional form, normality, and heteroscedasticity, based on the Lagrange Multiplier (LM) Test Statistics as shown in Table 12. In addition, an inspection of the cumulative sum (CUSUM) and the cumulative sum of squares (CUSUMSQ) graphs (Figures 1–6) from

the recursive estimation of the model reveals that there is stability and no systematic change detected in the coefficient at 5% significant level over the sample period.

**Table 12.** ARDL-UECM models diagnostic tests.

| LM Test Statistics Results | Model 1 | Model 2 | Model 3 |
|---|---|---|---|
| *R*-Square | 94.30% | 97.30% | 96.70% |
| Serial Correlation: CHSQ(1) | 0.159 (0.690) | 0.565 (0.452) | 0.009 (0.923) |
| Functional Form: CHSQ(1) | 1.136 (0.287) | 0.477 (0.490) | 0.265 (0.607) |
| Normality: CHSQ(2) | 2.782 (0.249) | 1.377 (0.502) | 0.664 (0.717) |
| Heteroscedasticity: CHSQ(1) | 0.124 (0.725) | 0.495 (0.482) | 0.147 (0.702) |

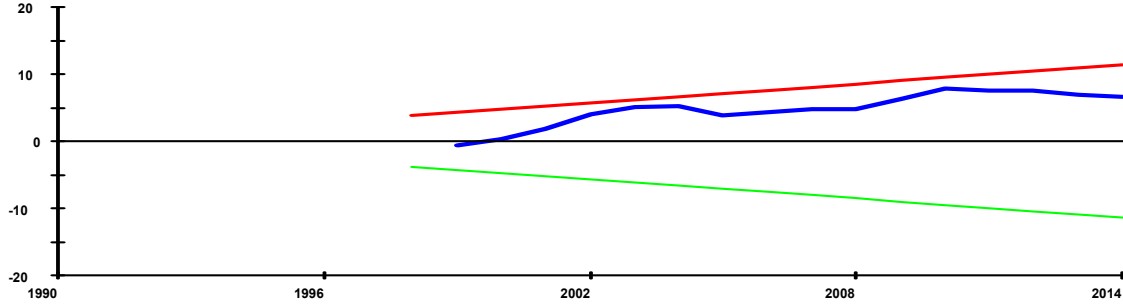

The straight line represents critical bounds at 5% significance level.

**Figure 1.** Plot of CUSUM (Cumulative Sum of Recursive Residuals) for coefficient stability for ECM model 1.

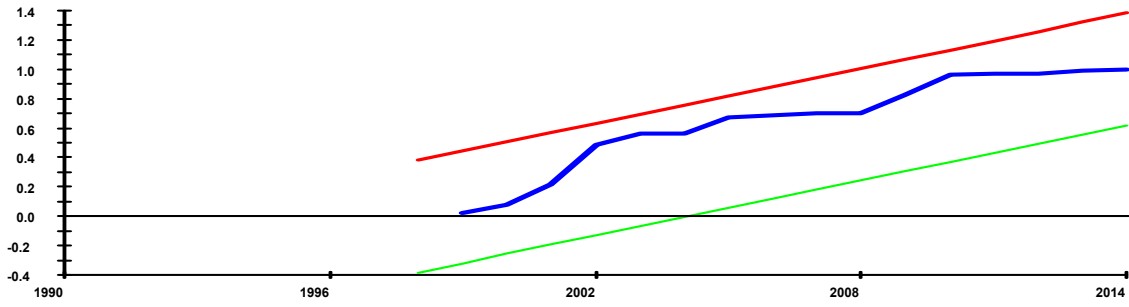

The straight line represents critical bounds at 5% significance level.

**Figure 2.** Plot of CUSUMSQ (Cumulative Sum of Squares of Recursive Residuals) for coefficient stability for ECM model 1.

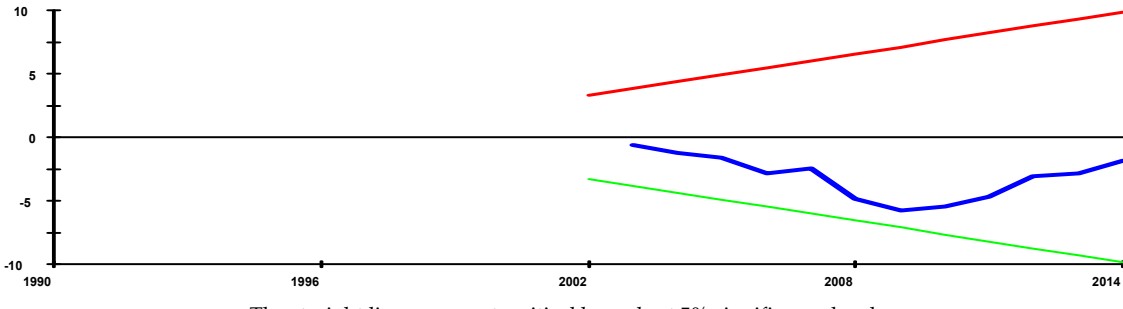

The straight line represents critical bounds at 5% significance level.

**Figure 3.** Plot of CUSUM (Cumulative Sum of Recursive Residuals) for coefficient stability for ECM model 2.

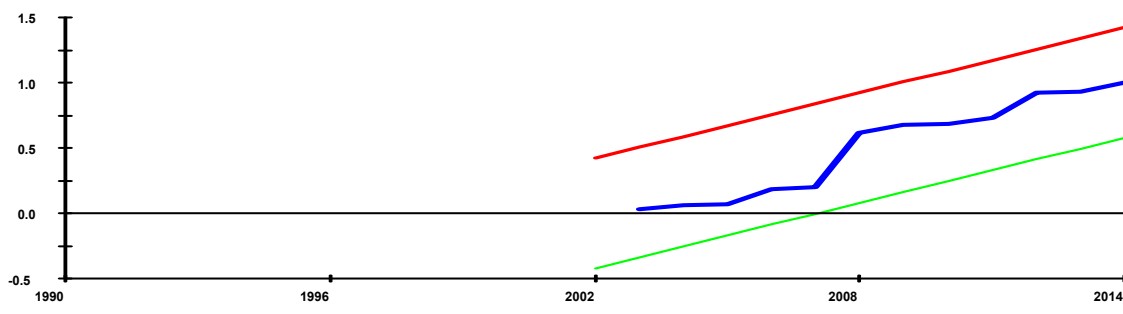

The straight line represents critical bounds at 5% significance level.

**Figure 4.** Plot of CUSUMSQ (Cumulative Sum of Squares of Recursive Residuals) for coefficient stability for ECM model 2.

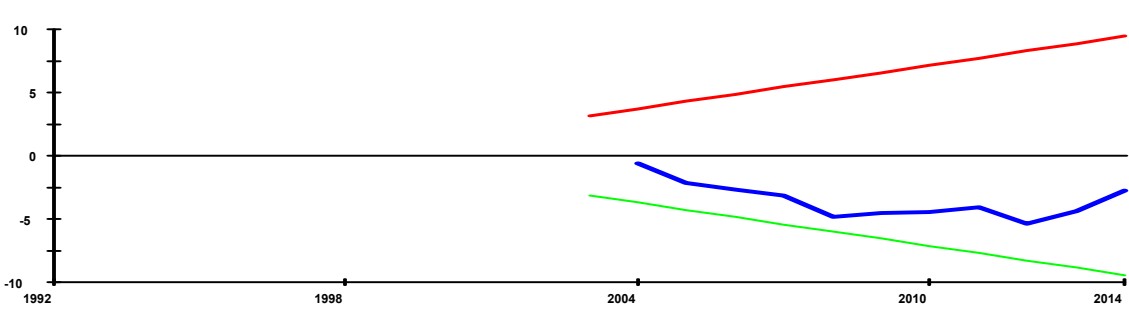

The straight line represents critical bounds at 5% significance level.

**Figure 5.** Plot of CUSUM (Cumulative Sum of Recursive Residuals) for coefficient stability for ECM model 3.

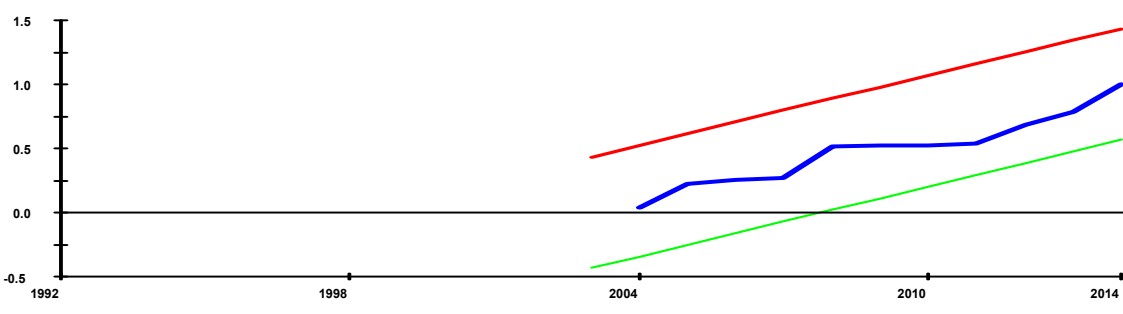

The straight line represents critical bounds at 5% significance level.

**Figure 6.** Plot of CUSUMSQ (Cumulative Sum of Squares of Recursive Residuals) for coefficient stability for ECM model 3.

## 5. Conclusions

The aim of this paper has been to empirically examine the impact of stock market evolution on economic growth in Nigeria. The paper employs the ARDL-bounds testing approach and unrestricted error correction model (UECM) to examine this impact. The paper also employs three proxies of stock market development, namely stock market capitalisation, stock market traded value and stock market turnover, together with a combined stock market development index for this analysis. According to Pesaran and Shin (1999 [28]), the ARDL methodology has robust properties in small sample sizes compared with traditional co-integration methodologies, which normally require a large sample size. In addition, Narayan (2004 [29]) shows that the ARDL method removes the uncertainty that comes with pretesting the order of integration of the variables.

The results of this study suggest that stock market development has a mixed effect on sustainable economic growth in Nigeria. Moreover, when the combined stock market evolution index is used,

stock market developments have a negative effect on sustainable economic growth in Nigeria. This may be due to the illiquid nature of the stock market in the country, as well as the magnitude of the sector compared to the economy. This finding is in contradiction to our expectation but supports the conclusions reached by some of the previous studies. In fact, another surprise finding of this paper shows that foreign direct investment also has a mixed impact on economic growth in Nigeria.

The policy implications arising out of the empirical findings are that the stock markets developments have been supportive, but more needs to be done to realise its full potential effects on economic growth in Nigeria. They can do this by increasing the financial deepening and the removal of bottlenecks in the financial sectors of the economy by providing further public education on the value of stock markets, improving public sector surveillance, abiding by stringent accounting standards and auditing best practices, and adopting proper legal framework to help shape the financial deepening process.

**Conflicts of Interest:** The author declares no conflict of interest.

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
