# Peer review of "Stock Market and Sustainable Economic Growth in Nigeria"

_economies, doi:10.3390/economies4040025_

Reviewer 1 Report

Report on “Stock Market and Sustainable Economic Growth in Nigeria”

Manuscript Number: economies-148088

The assessment of stock market impact on economic growth is far from new in the economics literature. Even though, the huge dimension of the financial problems faced by developing economies and potential effect of undeveloped financial markets on economic growth assure that it remains a pertinent topic in the economics field, and as so deserves further research. Nevertheless, in my opinion, the manuscript requires improvement to be a piece of useful research. In order to facilitate the author's task in achieving a publishable manuscript, I present the following considerations.

The main issues are:

1.      Overall, the manuscript lacks the structure of an empirical article. By large it is nebulous, repetitive, and quite verbose. Focus, as well as synthesis and conciseness are required to achieve a minimum of readability. The manuscript must be purged of what is established knowledge. For example, some sentences are not unlike those of a manual.

2.      The title of the manuscript is misleading; the “Sustainable” is not empirically researched.

3.      The manuscript has a notorious lack of focus. In fact, the authors consider too many issues. The empirical analysis would gain if it only focused on one model. For example, instead of working upon three models that include stock market related variables, why not center the manuscript on trying to assess which stock market variable is the most important to stimulate the economic growth.

4.      The manuscript would improve by incorporating essential information that would let other researchers replicate the empirical content of the manuscript. For example, for each variable should be indicated the source and the description of the raw data. To assure the full understand of econometric outputs, revealing the descriptive statistics is mandatory.

5.      The “Overview of stock market developments in Nigeria” section is quite wordy and it will benefit with synthesis and conciseness. Given that the time under research is from 1987 to 2014 it is difficult to perceive the interest of an analysis of the early 1960s.

6.      The “Methodology” suffers of inconsistency. The models 1 to 3 (lines 166-8) are of different nature than the models presented in lines 237-45. The first specification is not an Autoregressive Distributive Lag, therefore cannot be transformed in later one. The models must be numbered sequentially.

7.      The authors must justify why the study covers the period from 1987 to 2014. The number of degrees of freedom are too scarce to assure acceptable estimation of parameters statistical significance. Please, consider extending the time spam.

8.      Why do the authors only use the Phillips and Peron unit root tests? At least the authors must include the ADF tests and the confirmatory stationary KPSS test.

9.      The manuscript must provide the selection criteria used to select the ARDL specification used, i.e. “no intercept and no trend” or “restricted intercept and no trend”, and so on.

10.  Throughout Tables 4 to 9 there are plenty of variables not statistically significant. As the empirical research is a final stage of research, this kind of results should be absent or minimum. Please, reconsider the relevance of revealing these intermediate steps of the research.

Some minor issues:

1.        The equations must be numbered.

2.        It is strongly recommended to test for the presence of multicollinearity.

3.        In line 246 the authors state “φi = long run multipliers corresponding to long run relationships”. This is not true! Please, make the required transformations to make the sentence true.

4.        The methodology proposed in the lines 254-8, was not used, was it?

5.        In Table 3, the critical values correspond to “Table CI(iv) Case IV: Unrestricted intercept and restricted trend.” (Pesaran et al., 2001:301). Unfortunately, this does not match with the models the authors said to have tested!!!

6.        In the title of Table 10 the authors refer to “ARDL-VEC”. Where is the VEC?

7.        Some care is required in writing the references. For example, the reference of Owusu and Odhiambo (lines 417-8) is of 2014 not 2013.

8.        The econometric software used ought to be disclosed.

Author Response

Reviewer 1

Main Issues:

Point 1: We have removed some paragraphs from section 3 to make the manuscript more succinct.

Point 2: The word “sustainable” was included in the title because, the empirical results show that, in the short run, stock market developments have impact on economic growth. However, in the long run, there is no impact in Nigeria. This therefore, justifies the inclusion of the word in the title.

Point 3:  Just using one model will not allow us to fully investigate the impacts of stock market developments on economic growth in Nigeria. Using all the models allow us to make a robust conclusion about the relationship.

Point 4: All the data used in the paper are from the same sources which have been shown in section 4 under “Methodology and Empirical Analysis”

Point 5: Some paragraphs have been deleted to make the paper less wordy. The reasons for using data from only 1987 to 2014 have been given in the “Introduction”.

Point 6: Equations 1 to 3 are the long term relationship corresponding to the ARDL models in equation 4 to 6 (see also Owusu, 2012).

Point 7: Justification of the use of data from only 1987 is given in the introduction. Furthermore, Pesaran and Shin (1999) and Narayan (2004) also showed that, the ARDL method has robust parameters in small samples.

Point 8: We have included the results from ADF as well (see the revised paper, tables 1 and 2).

Point 9: The criteria for the selection of the ARDL models are indicated on top of results tables (See tables 6 to 11).

Point 10: The results shown are based on the models of best-fit selected on “general to specific” approach.

Some minor issues:

Point 1: The equations are now numbered

Point 2: The ARDL software used tests automatically for multicollinearity. If there is any multicollinear between the variables, the equation will not be estimated.

Point 3: Sentence amended.

Point 4: They were used. See results in table 12.

Point 5: The test was conducted at critical values where k=3 after “general to specific” best-fit approach.

Point 6: Corrected. It should have been “UECM”. See table 12.

Point 7: The paper was published online in November 2013 and then included in Volume 21 issue 4 in Jan 2014. However, I have changed to 2014.

Point 8: Included. See section 4 under “Methodology and Empirical Analysis”.

Reviewer 2 Report

This paper does a good job with examining the relationship between stock market evolution and sustainable economic growth in Nigeria. The study employs Auto‐Regressive Distributed Lag (ARDL)‐bounds testing approach and a combined stock market indicators index to examine the relationship. The main finding of the paper is that in the long run, stock markets have no positive and at best mixed effect on economic growth in Nigeria. This finding supports the numerous past studies, which have reported negative/mixed or inconclusive results on the effects of stock markets on economic growth. Another major contribution of this paper is that there is the need to increase financial deepening and the removal of bottlenecks in the financial sectors of the economy by providing further public and institutional education on the value of stock markets to economic development. The paper has a thorough literature review and a unique empirical literature review to provide readers with the contribution that this paper is making to the literature.  The methodology was sound and the conclusion was strong.